# Bovine Parainfluenza Virus Type 3 (BPIV3) Enters HeLa Cells via Clathrin-Mediated Endocytosis in a Cholesterol- and Dynamin-Dependent Manner

**DOI:** 10.3390/v13061035

**Published:** 2021-05-31

**Authors:** Wei Pan, Hui Nie, Hongmei Wang, Hongbin He

**Affiliations:** 1Ruminant Diseases Research Center, College of Life Sciences, Shandong Normal University, Jinan 250014, China; panwei662348@163.com (W.P.); hongmeiwang@sdnu.edu.cn (H.W.); 2Key Laboratory of Animal Resistant Biology of Shandong, College of Life Sciences, Shandong Normal University, Jinan 250014, China; niehui@sdnu.edu.cn

**Keywords:** BPIV3, virus entry, endocytosis, clathrin, cathepsins

## Abstract

Bovine parainfluenza virus 3 (BPIV3) is a crucial causative agent of respiratory disease in young and adult cattle. No specific therapies are available for BPIV3 infection. Understanding the internalization pathway of the virus will provide a new strategy for the development of antiviral therapy. Here, the mechanism of BPIV3 entry into HeLa cells was analyzed using RNA silencing and pharmacological inhibitors. Treatment of HeLa cells with hypertonic medium prevented BPIV3 internalization. These results indicated that BPIV3 entered HeLa cells via receptor-mediated endocytosis. Moreover, removing cell membrane cholesterol through MβCD treatment hampered viral penetration but not viral replication. In addition, BPIV3 infection was inhibited by pretreatment with dynasore or chlorpromazine (CPZ) or knockdown of dynamin II or clathrin heavy chain. However, virus entry was unaffected by nystatin, EIPA, wortmannin, or cytochalasin D treatment or caveolin-1 knockdown. These data demonstrated that the entry of BPIV3 into HeLa cells was dependent on clathrin-mediated endocytosis but not on caveolae-mediated endocytosis or the macropinocytosis pathway. Many viruses are transported to endosomes, which provide an acidic environment and release their genome upon separation from primary endocytic vesicles. However, we found that BPIV3 infection required endosomal cathepsins, but not a low pH. In summary, we show, for the first time, that BPIV3 enters HeLa cells through the clathrin-mediated endocytosis pathway, presenting novel insights into the invasion mechanism of Paramyxoviridae.

## 1. Introduction

Bovine parainfluenza virus type 3 (BPIV3), a member of the Paramyxoviridae family, is one of the most critical pathogens correlated to bovine respiratory disease complex (BRDC) in cattle [1]. BPIV3 is a nonsegmented negative-strand RNA virus that encodes six structural proteins, namely, hemagglutinin neuraminidase protein (HN), fusion protein (F), matrix protein (M), nuclear protein (N), phosphoprotein (P), large polymerase unit (L), and three nonstructural proteins: C, D, and V proteins [2]. To date, no specific treatment for BPIV3 infection is available. Elucidating the mechanisms by which BPIV3 enters host cells will provide potential therapeutic targets.

Viruses can utilize multiple endocytic pathways to penetrate cells. These main routes of endocytosis include clathrin-mediated endocytosis (CME), caveolae-mediated endocytosis (CavME), macropinocytosis, and clathrin- and caveolin-independent endocytosis [3]. Of these pathways, CME is the major entry route for the internalization of many viruses, such as vesicular stomatitis virus (VSV), bovine ephemeral fever virus (BEFV), African swine fever virus (ASFV), rabies virus (RABV), and porcine hemagglutinating encephalomyelitis virus (PHEV) [4,5,6,7,8]. Once internalization signals are activated upon ligand binding to specific receptors, cytoplasmic clathrin molecules are recruited to the plasma membrane and form clathrin-coated pits (CCPs). Facilitated by dynamin GTPase, CCPs separate from the plasma membrane and form mature vesicles coated with clathrin [9]. Another intensively studied pathway is caveolae-mediated endocytosis, which is characterized by the presence of caveolin [10]. Simian virus 40 (SV40) was the first pathogen reported to enter cells via the caveolae-dependent endocytosis pathway. Subsequently, other viruses, including echovirus 1 (EV1), human coronavirus 229E (HCoV-229E), amphotropic murine leukemia virus (A-MLV), and classical swine fever virus (CSFV) were reported to penetrate host cells through this endocytic pathway [11,12,13,14,15]. Due to the smallness of the vesicles that are endocytosed in association with clathrin or caveolin-1, some large viral particles undergo cellular internalization through other mechanisms, such as macropinocytosis. Macropinocytosis is undertaken by large endocytic vesicles formed at specific locations in the plasma membrane [16]. Macropinocytosis has been identified as an entry mechanism for several viruses, including vaccinia virus, Ebola virus, and adenovirus serotype 3 [17,18,19]. However, the detailed mechanism by which BPIV3 enters HeLa cells remains obscure. In this study, we explored the cell entry mechanisms of BPIV3 by using pharmacological inhibitors and RNA silencing. The results demonstrated that BPIV3 enters HeLa cells by following a cholesterol-, dynamin-, or clathrin-dependent pathway and requires cathepsins.

## 2. Materials and Methods

### 2.1. Cell Lines and Virus

Hela cells purchased from the Cell Bank of Type Culture Collection of the Chinese Academy of Sciences (Shanghai, China) were cultured in DMEM supplemented with 10% FBS and 2 mM L-glutamine. The cells passaged to the 10th generation were used for further experiments. BPIV3 (No. CGMCC9992) and VSV-IND (GenBank accession no. AM690336.1) were propagated in HeLa cells.

### 2.2. Inhibitors, Antibodies, and Reagents

Ammonium chloride (NH_4_Cl), chloroquine (CQ), dynasore, and cytochalasin D (Cyto D) were purchased from Sigma-Aldrich (St. Louis, MO, USA), and 5-(N-ethyl-N-isopropyl) amiloride (EIPA), nystatin, Cat Bi (Cathepsin B inhibitor, CA-074Me), chlorpromazine (CPZ), Cat Li (3-epiursolic acid), and methyl-β-cyclodextrin (MβCD) were purchased from MCE Corporation (New Jersey, USA). Antibodies against dynamin II, clathrin heavy chain (CHC), and caveolin-1 were obtained from Cell Signaling Technology (Danvers, MA, USA). The rabbit polyclonal antibody against the BPIV3 HN protein was prepared in our laboratory. The rabbit monoclonal antibody against vesicular stomatitis virus (VSV) G protein was purchased from Abways Technology (Shanghai, China). Alexa Fluor 555-conjugated cholera toxin subunit B (CTB-AF555) was purchased from Invitrogen (Carlsbad, CA, USA).

### 2.3. Inhibitor Administration and Cell Viability Assay

HeLa cells cultured in 96-well plates were treated with various concentrations of pharmacological inhibitors for 7 h, except for MβCD, CPZ, wortmannin, CQ, and NH_4_Cl, which were incubated with the cells for 3 h. After washing three times with PBS, 100 μL of medium containing 10 μL of CCK-8 solution was added to each well and cultured at 37 °C for 1 h, then the absorbance was evaluated at 450 nm with a microplate reader (China elisa microplate reader DNM-9602A).

### 2.4. Virus Internalization Assay

To detect virus entry, HeLa cells were cultured with medium containing indicated concentrations of chemical inhibitors (dynasore, Cyto D, EIPA, nystatin, Cat Bi, and Cat Li) for 1 h and infected with BPIV3 at 4 °C for 1 h. Then, the unbound viruses were removed by washing with ice-cold PBS three times, and the cells were shifted to a 37 °C environment to maintain inhibitor action. After the culture for the designated time, the cells were lysed and subjected to western blot or qRT-PCR analysis. For MβCD, CPZ, wortmannin, CQ, and NH_4_Cl, HeLa cells were cultured with medium containing indicated concentrations of chemical inhibitors for 1 h and infected with BPIV3 for 1 h at 4 °C, followed by incubation for 1 h at 37 °C. Then, the noninternalized viruses were removed with citrate buffer (pH 3.1). After the cells were cultured without inhibitors for the designated time, they were washed with PBS three times, and total protein was extracted using RIPA lysis buffer supplemented with 1 mM PMSF. Proteins of the different groups were subjected to western blot or qRT-PCR analysis.

### 2.5. siRNA Transfection

Briefly, HeLa cells seeded in 6-well plates were transfected by adding 2 mL of maintenance medium with the indicated concentration of siRNA and 6 µL of Attractene transfection reagent (Qiagen, Germantown, MD, USA) to the culture. The transfection medium was removed and replaced with DMEM containing 2% FBS, and the cells were cultured for another 24 h before inoculation. The siRNA sequences used are shown in Table 1.

### 2.6. Western Blot and qRT-PCR Analyses

Total protein was extracted using RIPA lysis buffer, separated on 12% SDS-PAGE gels, and then blotted onto PVDF membranes (Millipore, Burlington, MA, USA). After blocking for 1 h with blocking buffer, the membranes were incubated overnight with primary antibodies diluted in 0.1% Tween-20/PBS (PBST) at 4 °C. After being washed with TBST, the membranes were incubated with secondary antibody at 37 °C for 1 h. Immunodetection was performed with an ECL detection kit (Thermo Fisher Scientific, Waltham, MA, USA).

Total RNA was extracted from cells using a cell total RNA isolation kit (Foregene Biotech, Chengdu, China). The RNA was quantified using a NanoDrop spectrophotometer (Thermo Fisher Scientific, Waltham, MA, USA) and transcribed into complementary DNA using M-MLV reverse transcriptase (catalog No. 2641A; TaKaRa, Dalian, China). Oligo (dT) and BPIV3F primers were used to prime cDNA synthesis. The qRT-PCR assay was performed using SYBR Green PCR Master Mix (TaKaRa, Dalian, China). The primer sequences used are shown in Table 2.

### 2.7. Statistical Analysis

The statistical significance was analyzed using ANOVA for multiple comparisons. The results are expressed as the average ± standard deviation (SD) obtained from three independent experiments.

## 3. Results

### 3.1. Inhibition of Endocytic Uptake from the Cell Surface Inhibits BPIV3 Entry into HeLa Cells

To evaluate the role of cholesterol in BPIV3 infection, MβCD was used to deplete cholesterol in the cell membrane. First, HeLa cells were incubated with MβCD for 3 h, and the cytotoxic effects of the inhibitor were assessed. The cells could tolerate at least 4 mM MβCD (Figure 1A). Second, to test whether depletion of cholesterol affected viral internalization, HeLa cells were preincubated with various concentrations of MβCD prior to virus inoculation. Compared with that in the mock incubation group, the level of viral HN protein in the MβCD-treated cells was significantly decreased (Figure 1B). To preclude the possibility that MβCD may affect BPIV3 replication, HeLa cells were treated with increasing concentrations of MβCD 1 h postinfection with BPIV3. The results showed that post-treatment of these cells with MβCD had no effect on the expression of the viral protein (Figure 1C). To confirm these results, the viral genome was detected by RT-qPCR 2 h postinfection with BPIV3. A significant reduction in the viral genome in the pretreated but not the post-treated cells was observed (Figure 1D). Altogether, these findings indicated that cholesterol was also required for viral internalization by HeLa cells.

To determine whether BPIV3 enters cells via an endocytic mechanism during infection, HeLa cells were pretreated with hypertonic medium, which inhibits receptor-mediated endocytosis, and infected with BPIV3 [20]. Although no cytotoxic effect was observed in HeLa cells cultured with 0.3 M sucrose, the BPIV3 protein level was significantly decreased compared with that in the control groups (Figure 1E,F). To preclude the possibility that the hypertonic medium interfered with viral replication, the viral genome was detected by RT-qPCR 2 h postinfection with BPIV3. The results showed that these inhibitors significantly reduced virus entry (Figure 1G). Taken together, these results suggest that BPIV3 efficiently enters HeLa cells through an endocytic mechanism.

### 3.2. Entry of BPIV3 into HeLa Cells Is Dynamin-Dependent

To assess whether BPIV3 entry is dynamin-dependent, dynasore was used to confirm the role of dynamin in BPIV3 entry. HeLa cells were preincubated with dynasore before viral infection. As shown in Figure 2A, the viability remained unchanged for cells incubated with up to 50 μM dynasore. Due to the dynamin-dependent endocytic mechanism of VSV, we used VSV-infected cells as positive controls to test the effectiveness of dynasore. The results showed that treatment with dynasore reduced VSV G protein expression levels, suggesting that dynasore has an effect in HeLa cells (Figure 2B). Then, we determined the effect of dynasore on BPIV3 infection. The results showed that the levels of the BPIV3 HN protein and the viral genome decreased as the concentration of dynasore was increased (Figure 2C,D). To validate this result, we measured the effect of dynamin II silencing on BPIV3 infection. The results showed that BPIV3 HN protein expression was reduced upon dynamin II knockdown (Figure 2E). Taken together, these results demonstrated that BPIV3 entered HeLa cells in a dynamin-dependent manner.

### 3.3. Clathrin-Mediated Endocytosis Is Involved in BPIV3 Entry into HeLa Cells

To test whether clathrin-mediated endocytosis is involved in viral invasion, HeLa cells were treated with chlorpromazine (CPZ), an inhibitor of clathrin-coated pit formation. The cytotoxic effects were investigated with a cell viability assay, and cells infected with VSV, which is dependent on clathrin-mediated endocytosis, served as positive controls (Figure 3A,B). After determining the subtoxic dose and effectiveness of CPZ, we determined the effect of CPZ on BPIV3 entry. As shown in Figure 3C,D, the expression level of the BPIV3 HN protein and viral genome decreased in the CPZ-treated cells in a dose-dependent manner. We also investigated the role of clathrin in BPIV3 internalization by knocking down the clathrin heavy chain (CHC) level. Compared with the control siRNA-transfected cells, reducing CHC expression decreased the level of the BPIV3 HN protein (Figure 3E). Taken together, these results demonstrated that BPIV3 penetration into HeLa cells was clathrin-dependent.

### 3.4. Entry of BPIV3 into HeLa Cells Is Caveolin-Independent

To investigate whether caveolae-mediated endocytosis pathways are involved in BPIV3 for entry, HeLa cells were first incubated with nystatin, a well-known caveolae-mediated endocytosis inhibitor. The toxicity of nystatin was determined, and as shown in Figure 4A, no cytotoxic effect was observed in HeLa cells. To test the efficacy of nystatin, the uptake of Alexa Fluor 555-labeled cholera toxin subunit B (CTB) was examined. As shown in Appendix A, the internalization of CTB was decreased in the cells pretreated with 20 to 50 µM nystatin. However, neither the BPIV3 HN protein nor the viral genome level was changed in the cells treated with different concentrations of nystatin (Figure 4B,C). These results were confirmed by siRNA knockdown of caveolin-1, which is crucial for the formation of stable caveolin-coated vesicles. Western blotting showed that the expression of caveolin-1 was decreased when cells were transfected with increasing concentrations of siRNA targeting caveolin-1. However, the expression of the BPIV3 HN protein in the different groups was not changed (Figure 4D). Taken together, these results demonstrated that the entry of BPIV3 into HeLa cells was caveolin-independent.

### 3.5. Entry of BPIV3 into HeLa Cells Is Macropinocytosis-Independent

We next investigated the role of macropinocytosis in BPIV3 internalization. First, the cytotoxic effects of macropinocytosis inhibitors were determined. As shown in Figure 5A, the inhibitors used in this study had no effect on cell viability. As a potent inhibitor of the Na^+^/H^+^ exchanger, EIPA has been shown to block macropinocytosis formation [21]. Therefore, we first used EIPA to explore the role of macropinocytosis in the cellular entry of BPIV3. However, the level of the BPIV3 HN protein was not changed in the cells treated with different concentrations of EIPA (Figure 5B). Macropinocytosis was shown to rely on phosphatidylinositol 3-kinase (PI3K) activation [22]. Therefore, wortmannin, an inhibitor of this kinase, was used to confirm the result. As shown in Figure 5C, inhibition of PI3K activity with various concentrations of wortmannin did not inhibit BPIV3 HN protein synthesis. Polymerization of F-actin was shown to be essential for the formation of membrane protrusions during micropinocytosis [23]. The results of our study were confirmed by using the actin polymerization-specific inhibitor cytochalasin D (Cyto D), which blocks micropinocytosis-mediated endocytosis. Compared with previous results, the entry of BPIV3 was not inhibited when HeLa cells were treated with cyto D (Figure 5D). Taken together, these results suggested that viral invasion was macropinocytosis-independent.

### 3.6. Entry of BPIV3 into HeLa Cells Is pH-Independent

To define the role of pH in the viral invasion, we first investigated the effects of two acidification inhibitors, NH_4_Cl and chloroquine (CQ), on BPIV3 infection. Subtoxic doses of these inhibitors were determined by performing a cell viability assay. As shown in Figure 6A, no dose of the inhibitors used in this experiment had a cytotoxic effect on the HeLa cells. HeLa cells were preincubated with increasing concentrations of NH_4_Cl and chloroquine (CQ) for 1 h, followed by infection with BPIV3 at an MOI of 1. As shown in Figure 6C,E, neither NH_4_Cl nor CQ inhibited the expression of the BPIV3 HN protein, although both agents potently inhibited the entry of vesicular stomatitis virus, which required endosomal acidification for its fusion and entry (Figure 6B,D). These results were further confirmed by detecting the genome level of BPIV3 (Figure 6F). In accordance with previous results, there was no obvious elevation in the viral genome level in the cells treated with different concentrations of NH_4_Cl or CQ. Taken together, these results demonstrated that BPIV3 penetrated HeLa cells in a pH-independent manner.

### 3.7. BPIV3 Infection of HeLa Cells Requires Cathepsin Activity

To examine the possible role of cathepsin B and L in BPIV3 infection, HeLa cells were incubated with increasing concentrations of cathepsin B and L inhibitors and then infected with BPIV3. As shown in Figure 7A,B, no dose of the inhibitors used in this experiment had a cytotoxic effect on the HeLa cells. However, the level of HN was remarkably reduced in both cathepsin B and L inhibitor-treated cells (Figure 7C,D). These results indicated that cathepsins B and L played important roles in BPIV3 infection. To further determine the roles of cathepsin B and L on BPIV3 entry, the viral genome level in cathepsin B and L inhibitor pretreated cells was detected by RT-qPCR at 2 h postinfection with BPIV3. The results showed that these inhibitors significantly reduced virus entry (Figure 8E,F). These results indicated that the entry of BPIV3 into HeLa cells required cathepsins.

## 4. Discussion

Many *paramyxoviruses* enter host cells through direct fusion of the viral envelope with the cell membrane and the subsequent release of viral genome into the cytoplasm, a process that explains a nonendocytic mechanism [24]. To determine whether BPIV3 enters HeLa cells via an endocytic or nonendocytic mechanism, hypertonic medium containing 0.3 M sucrose was used in experiments, and the results demonstrated that BPIV3 efficiently enters HeLa cells via an endocytic mechanism.

The process of virus penetration of target cells occurs in a stepwise manner that involves several cellular factors. As a cellular bioprocess regulatory factor, cholesterol was reported to participate in many viral internalizations [25]. A previous study demonstrated that cholesterol was essential for BPIV3 entry into MDBK cells [26]. However, the role of cholesterol in BPIV3 entry into HeLa cells remained unelucidated. Here, we demonstrated that depletion of cholesterol through MβCD treatment inhibited BPIV3 entry but not BPIV3 replication in HeLa cells. This evidence suggests that extraction of cholesterol with MβCD could impede clathrin- and caveolae-mediated endocytosis [27]. Then, we determined the role of dynamin in BPIV3 penetration of cells. Dynamin is a GTPase that plays a critical role in the fission of clathrin- and caveola-coated vesicle formation [28]. We demonstrated that the entry of BPIV3 into HeLa cells required dynamin. These results prompted us to investigate the role of clathrin- and caveolae-mediated endocytosis in virus entry. First, chlorpromazine and siRNA targeting clathrin heavy chain (CHC) were used to specifically repress clathrin-dependent endocytosis. Our experiments demonstrated that both chlorpromazine treatment and CHC knockdown inhibited the entry of BPIV3 into HeLa cells. These results indicated that clathrin-mediated endocytosis was important for viral infection. Then, we determined the role of caveolae-mediated endocytosis in BPIV3 entry by using nystatin and siRNA targeting caveolin-1 to inhibit caveolae-mediated endocytosis. The results showed that neither nystatin nor caveolin-1 knockdown inhibited BPIV3 infection of HeLa cells, although nystatin effectively blocked cholera toxin (CTB) uptake. These results indicated that BPIV3 did not enter HeLa cells through the caveolae-mediated pathway. In addition to the clathrin- and caveolae-dependent endocytic routes, macropinocytosis is a distinct endocytosis pathway that has gained growing attention due to its roles in virus entry and immune defense [23]. Here, we used three inhibitors targeting macropinocytosis to confirm its role in mediating viral infection. However, neither inhibitor effectively blocked BPIV3 entry into HeLa cells. These results indicated that BPIV3 was not internalized via macropinocytosis.

For the majority of viruses, acidification of the endosome can lead to the release of the viral genome into the cytosol. Thus, we tested the role of pH in BPIV3 infection and found that BPIV3 entry was pH-independent. In addition to providing an acidic environment, endosomes are rich in proteases. Endosomal cysteine proteases (primarily cathepsin L and B) were shown to be important for many viral infections, including severe acute respiratory syndrome (SARS) coronavirus, reovirus, Ebola virus, and murine hepatitis virus 2 infections [29,30,31,32]. Thus, we examined the roles of these cathepsins in viral infection and found that BPIV3 entry required cathepsin B and L. Cara demonstrated that Hendra virus, an important *paramyxovirus,* leverages the cellular protease cathepsin L to cleave the viral F protein, which elicits fusion activity [33]. The mechanism by which cathepsin L affects BPIV3 entry needs further study.

## 5. Conclusions

In summary, our results demonstrate that cholesterol also plays critical roles in BPIV3 entry HeLa cells. In contrast with some other Paramyxoviridae, endocytic mechanism could be utilized by BPIV3 to enter HeLa cells. Further investigations demonstrate that entry of BPIV3 into HeLa cells is dependent on clathrin- but not caveolae-dependent endocytosis or the macropinocytosis pathway. Trafficking through endosomes is known to be important for viruses that depend on low pH or endosomal cathepsin proteases to complete the entry process. However, we found that BPIV3 infection requires cathepsins but not low pH.

## Figures and Tables

**Figure 1 viruses-13-01035-f001:**
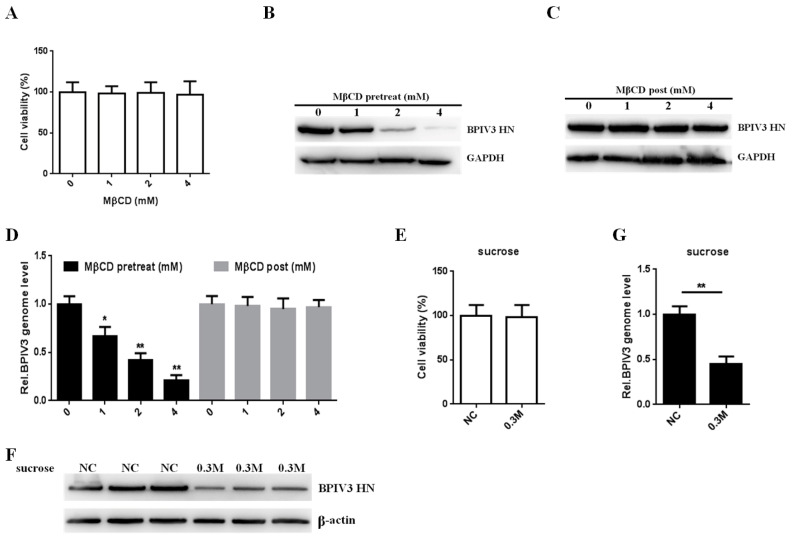
Effect of MβCD and hypertonic medium on BPIV3 infection. (**A**) The cytotoxicity of MβCD was determined by CCK-8 assays (0 vs. 1, *p* = 0.5743; 0 vs. 2, *p* = 0.6724; 0 vs. 4, *p* = 0.7215). (**B**) MβCD inhibited penetration of BPIV3. HeLa cells were preincubated with MβCD and infected with BPIV3 (MOI = 1). After the infected cells were incubated for 5 h, levels of BPIV3 HN were were determined by western blot. GAPDH served as the loading control. (**C**) MβCD had no effect on BPIV3 replication. HeLa cells preinoculated with BPIV3 (MOI = 1) for 1 h were treated with MβCD for 1 h. After the infected cells were incubated for 5 h, levels of BPIV3 HN were determined by western blot. GAPDH served as the loading control. (**D**) HeLa cells were pretreated or post-treated with MβCD and inoculated with BPIV3 (MOI = 1). After the infected cells were incubated for 2 h, the viral genome level was detected by RT-qPCR (for MβCD pretreated groups: 0 vs. 1, *p* = 0.014; 0 vs. 2, *p* = 0.0008; 0 vs. 4, *p* = 0.0002; for MβCD post-treated groups: 0 vs. 1, *p* = 0.7994; 0 vs. 2, *p* = 0.5654; 0 vs. 4, *p* = 0.6649). (**E**) The cytotoxicity of 0.3 M sucrose was determined by CCK-8 assays (*p* = 0.8601). (**F**) Hypertonic medium suppressed BPIV3 infection. HeLa cells were preincubated with medium containing either 0.3 M sucrose (hypertonic medium) or normal culture medium (NC) and infected with BPIV3 (MOI = 1). After the infected cells were incubated for 5 h, levels of BPIV3 HN were determined by western blot. β-actin served as the loading control. (**G**) Hypertonic medium inhibited BPIV3 entry. HeLa cells were pretreated with 0.3 M sucrose at the indicated concentrations and inoculated with BPIV3 (MOI = 1). After the infected cells were incubated for 2 h, the viral genome level was detected by RT-qPCR (*p* = 0.0015). The data are based on three independent experiments with at least three replicates for each. The data are shown as the means ± SD (** *p* < 0.01; * *p* < 0.05;).

**Figure 2 viruses-13-01035-f002:**
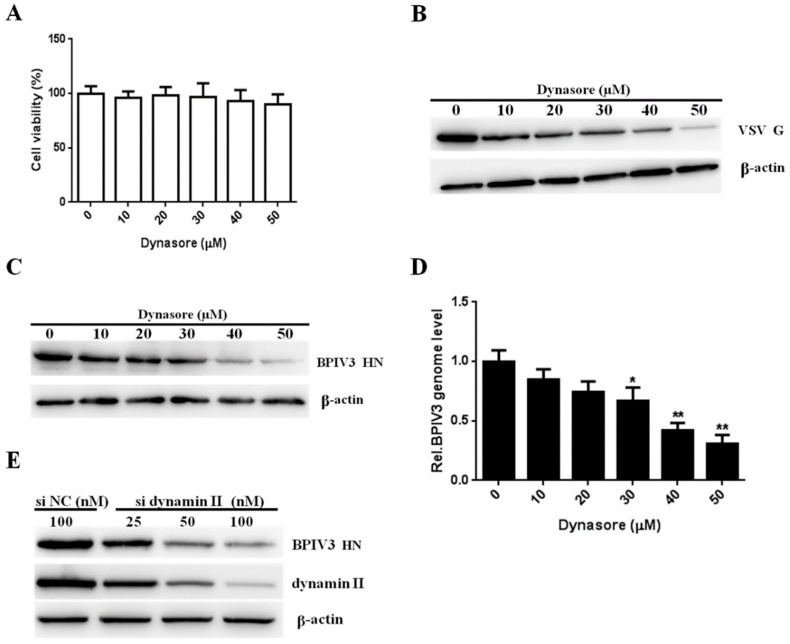
BPIV3 entry depends on dynamin. (**A**) The cytotoxicity of dynasore was determined by CCK-8 assays (0 vs. 10, *p* = 0.4780; 0 vs. 20, *p* = 0.7549; 0 vs. 30, *p* = 0.7295; 0 vs. 40, *p* = 0.3767; 0 vs. 50, *p* = 0.2053). (**B**,**C**) Dynasore inhibited VSV and BPIV3 infection. HeLa cells were pretreated with dynasore at the indicated concentrations and inoculated with VSV (MOI = 1) or BPIV3 (MOI = 1). After the infected cells were incubated for 4 h (for VSV) or 5 h (for BPIV3), levels of VSV G and BPIV3 HN were determined by western blot. β-actin served as the loading control. (**D**) HeLa cells were pretreated with dynasore at the indicated concentrations and inoculated with BPIV3 (MOI = 1) and incubated for 2 h. The viral genome level was determined by RT-qPCR (0 vs. 10, *p* = 0.4072; 0 vs. 20, *p* = 0.0823; 0 vs. 30, *p* = 0.0423; 0 vs. 40, *p* = 0.0076; 0 vs. 50, *p* = 0.0021). (**E**) HeLa cells transfected with the siRNA negative control (siNC) or increasing amounts of siRNA targeting dynamin II (si dynamin II) were inoculated with BPIV3 (MOI = 1) and incubated for 5 h, then the BPIV3 HN and dynamin II protein levels were determined by western blot. β-actin served as the loading control. The data are based on three independent experiments with at least three replicates for each. The data are displayed as the means ± SD (** *p* < 0.01; * *p* < 0.05;).

**Figure 3 viruses-13-01035-f003:**
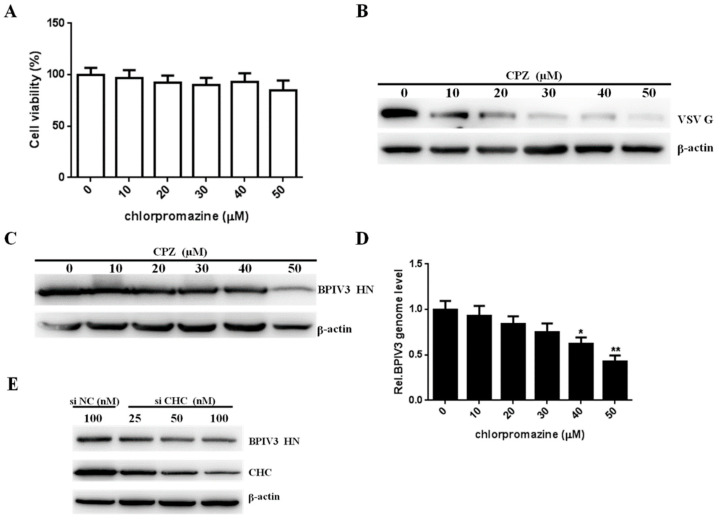
Clathrin is required for BPIV3 entry. (**A**) The cytotoxicity of CPZ was determined by CCK-8 assays (0 vs. 10, *p* = 0.6302; 0 vs. 20, *p* = 0.2345; 0 vs. 30, *p* = 0.1461; 0 vs. 40, *p* = 0.3222; 0 vs. 50, *p* = 0.0860). (**B**,**C**) CPZ inhibited VSV and BPIV3 infection. HeLa cells were preincubated with CPZ at the indicated concentrations and inoculated with VSV (MOI = 1) or BPIV3 (MOI = 1). After the infected cells were incubated for 4 h (for VSV) or 5 h (for BPIV3), levels of VSV G and BPIV3 HN were determined by western blot. β-actin served as the loading control. (**D**) HeLa cells preincubated with CPZ were infected with BPIV3 (MOI = 1) and incubated for 2 h, and the viral genome level was determined by RT-qPCR (0 vs. 10, *p* = 0.4473; 0 vs. 20, *p* = 0.0915; 0 vs. 30, *p* = 0.0612; 0 vs. 40, *p* = 0.0231; 0 vs. 50, *p* = 0.0072). (**E**) HeLa cells transfected with nontargeting control RNA (siNC) or increasing amounts of siRNA targeting CHC (siCHC) were infected with BPIV3 (MOI = 1) and incubated for 5 h, and the viral HN and CHC protein levels were determined by western blot analysis. β-actin served as the loading control. The data are based on three independent experiments with at least three replicates for each. The data are displayed as the means ± SD (** *p* < 0.01; * *p* < 0.05;).

**Figure 4 viruses-13-01035-f004:**
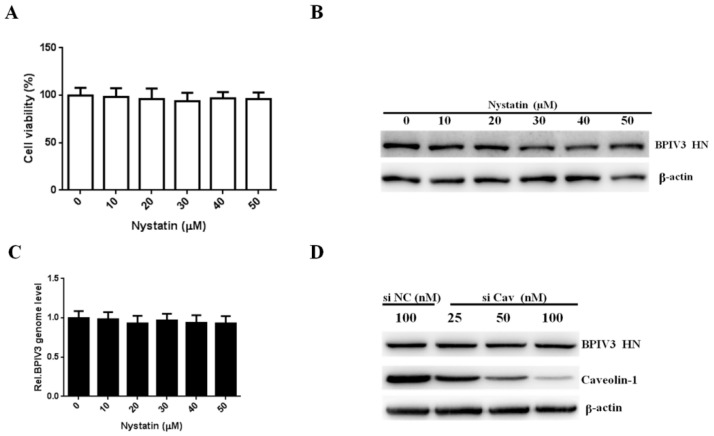
Caveolin is not required for BPIV3 entry. (**A**) The cytotoxicity of nystatin was determined by CCK-8 assays. (0 vs. 10, *p* = 0.7913; 0 vs. 20, *p* = 0.6386; 0 vs. 30, *p* = 0.4248; 0 vs. 40, *p* = 0.6319; 0 vs. 50, *p* = 0.5440). (**B**) HeLa cells were pretreated with nystatin and inoculated with BPIV3 (MOI = 1). After the infected cells were incubated for 5 h, levels of BPIV3 HN were determined by western blot. β-actin served as the loading control. (**C**) HeLa cells were pretreated with nystatin at the indicated concentrations and inoculated with BPIV3 (MOI = 1) for 2 h, and the viral genome level was determined by RT-qPCR (0 vs. 10, *p* = 0.7980; 0 vs. 20, *p* = 0.4020; 0 vs. 30, *p* = 0.6862; 0 vs. 40, *p* = 0.4580; 0 vs. 50, *p* = 0.3902). (**D**) HeLa cells transfected with siRNA negative control (siNC) or increasing amounts of siRNA targeting caveolin-1 (siCav) were infected with BPIV3 (MOI = 1) and incubated for 5 h, and viral HN and caveolin-1 protein levels were detected by western blot analysis. The data are based on three independent experiments with at least three replicates for each. The data are displayed as the means± SD.

**Figure 5 viruses-13-01035-f005:**
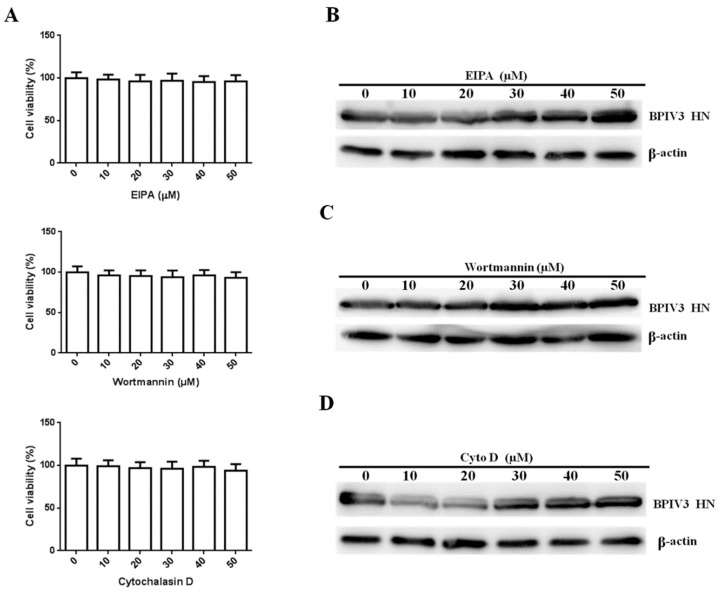
BPIV3 entry does not depend on macropinocytosis (for EIPA: 0 vs. 10, *p* = 0.7201; 0 vs. 20, *p* = 0.5405; 0 vs. 30, *p* = 0.6511; 0 vs. 40, *p* = 0.4288; 0 vs. 50, *p* = 0.5257; for wortmannin: 0 vs. 10, *p* = 0.5059; 0 vs. 20, *p* = 0.4400; 0 vs. 30, *p* = 0.3893; 0 vs. 40, *p* = 0.5171; 0 vs. 50, *p* = 0.2938; for cytochalasin D: 0 vs. 10, *p* = 0.8760; 0 vs. 20, *p* = 0.6423; 0 vs. 30, *p* = 0.5754; 0 vs. 40, *p* = 0.7631; 0 vs. 50, *p* = 0.3934). (**A**) Cell viability upon EIPA, wortmannin, and cytochalasin D treatment was evaluated. HeLa cells seeded in 96-well plates were treated with EIPA, cytochalasin D for 7 h, or wortmannin for 3 h at the indicated concentrations, then cell viability was determined as described in the text. (**B**–**D**) HeLa cells were pretreated with EIPA, wortmannin, or cytochalasin D at the indicated concentrations and inoculated with BPIV3 (MOI = 1). After the infected cells were incubated for 5 h, levels of BPIV3 HN were determined by western blot. β-actin served as the loading control. The data are based on three independent experiments with at least three replicates for each. The data are displayed as the means ± SD.

**Figure 6 viruses-13-01035-f006:**
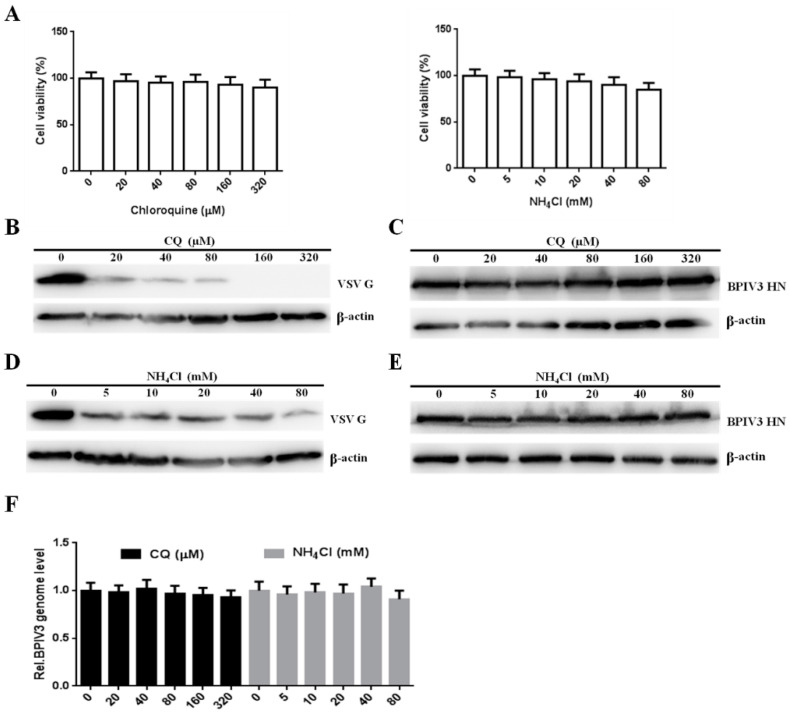
The entry of BPIV3 into HeLa cells is pH-independent. (**A**) Cell viability upon CQ or NH_4_Cl treatment was determined (for CQ: 0 vs. 20, *p* = 0.6227; 0 vs. 40, *p* = 0.4091; 0 vs. 80, *p* = 0.5350; 0 vs. 160, *p* = 0.3147; 0 vs. 320, *p* = 0.1833; for NH_4_Cl: 0 vs. 5, *p* = 0.7403; 0 vs. 10, *p* = 0.5028; 0 vs. 20, *p* = 0.3572; 0 vs. 40, *p* = 0.1802; 0 vs. 80, *p* = 0.0554). (**B**,**D**) HeLa cells preincubated with CQ or NH_4_Cl were infected with VSV (MOI = 1). After the infected cells were incubated for 4 h, levels of VSV G were determined by western blot. β-actin served as the loading control. (**C**,**E**) HeLa cells were pretreated with CQ or NH_4_Cl at the indicated concentrations and inoculated with BPIV3 (MOI = 1). After the infected cells were incubated for 5 h, levels of BPIV3 HN were determined by western blot. β-actin served as the loading control. (**F**) HeLa cells preincubated with CQ or NH_4_Cl were infected with BPIV3 (MOI = 1). The viral genome level was detected by RT-qPCR 2 h postinfection (for CQ: 0 vs. 20, *p* = 0.7710; 0 vs. 40, *p* = 0.7937; 0 vs. 80, *p* = 0.6773; 0 vs. 160, *p* = 0.4894; 0 vs. 320, *p* = 0.3318; for NH_4_Cl: 0 vs. 5, *p* = 0.6098; 0 vs. 10, *p* = 0.8042; 0 vs. 20, *p* = 0.7144; 0 vs. 40, *p* = 0.6153; 0 vs. 80, *p* = 0.2970). The data are based on three independent experiments with at least three replicates for each. The data are displayed as the means ± SD.

**Figure 7 viruses-13-01035-f007:**
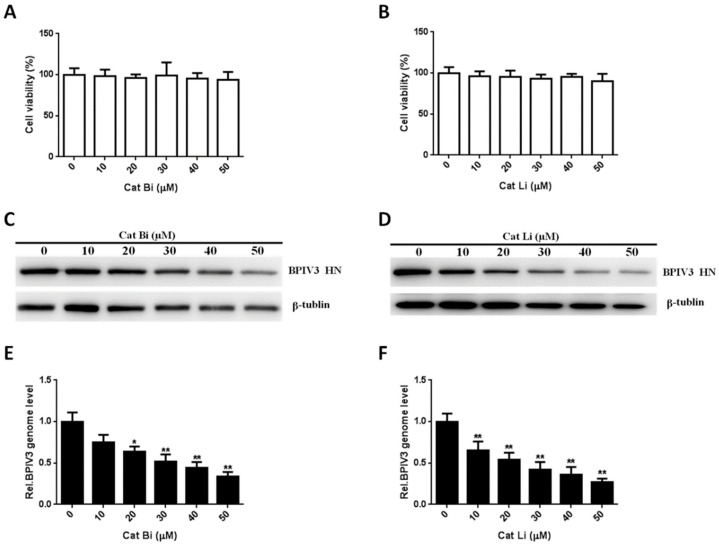
The entry of BPIV3 into HeLa cells requires cathepsin activity. (**A**,**B**) Cell viability upon Cat Bi or Cat Li treatment was determined, as described in the text (for Cat Bi: 0 vs. 10, *p* = 0.7748; 0 vs. 20, *p* = 0.4801; 0 vs. 30, *p* = 0.9271; 0 vs. 40, *p* = 0.4551; 0 vs. 50, *p* = 0.4402; for Cat Li: 0 vs. 10, *p* = 0.4942; 0 vs. 20, *p* = 0.4610; 0 vs. 30, *p* = 0.2315; 0 vs. 40, *p* = 0.3432; 0 vs. 50, *p* = 0.2034). (**C**,**D**) HeLa cells were pretreated with Cat Bi or Cat Li at the indicated concentrations and inoculated with BPIV3 (MOI = 1). After the infected cells were incubated for 5 h, levels of BPIV3 HN were determined by western blot. β-tublin served as the loading control. (**E**,**F**) HeLa cells were pretreated with Cat Bi or Cat Li at the indicated concentrations and inoculated with BPIV3 (MOI = 1). The infected cells were incubated for 2 h, and the viral genome level was detected by RT-qPCR (for Cat Bi: 0 vs. 10, *p* = 0.0782; 0 vs. 20, *p* = 0.0276; 0 vs. 30, *p* = 0.0039; 0 vs. 40, *p* = 0.0018; 0 vs. 50, *p* = 0.0007; for Cat Li: 0 vs. 10, *p* = 0.0087; 0 vs. 20, *p* = 0.0035; 0 vs. 30, *p* = 0.0017; 0 vs. 40, *p* = 0.0012; 0 vs. 50, *p* = 0.0003). The data are based on three independent experiments with at least three replicates for each. The data are displayed as the means ± SD (** *p* < 0.01; * *p* < 0.05).

**Figure 8 viruses-13-01035-f008:**
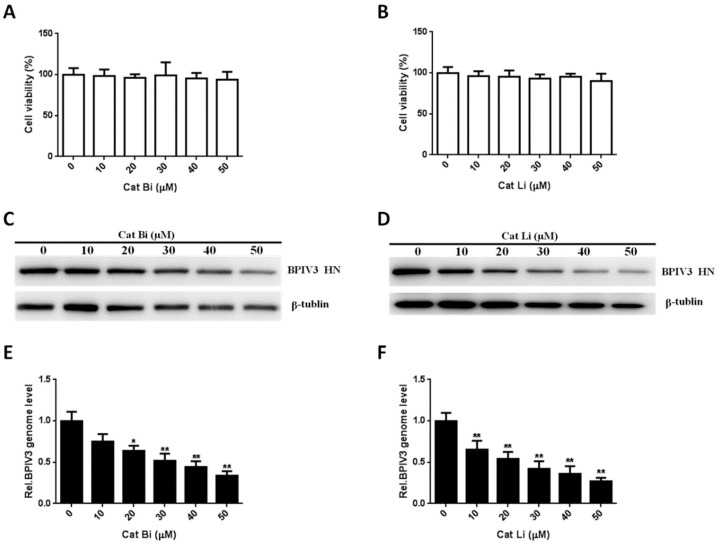
Entry of BPIV3 into HeLa cells requires cathepsin activity. (**A**,**B**) Cell viability upon Cat Bi or Cat Li treatment was determined, as described in the text. (**C**,**D**) HeLa cells were pretreated with Cat Bi or Cat Li at the indicated concentrations and inoculated with BPIV3 (MOI = 1). After the infected cells were incubated for 5 h, they were harvested and subjected to western blot analysis. (**E**,**F**) HeLa cells were pretreated with Cat Bi or Cat Li at the indicated concentrations and inoculated with BPIV3 (MOI = 1). The infected cells were incubated for 2 h, and the viral genome level was detected by RT-qPCR. The data are based on 3 independent experiments with at least three replicates for each. The data are displayed as the means± SD (** *p* < 0.01; * *p* < 0.05).

**Table 1 viruses-13-01035-t001:** siRNA sequences.

Gene Name	siRNA Name	siRNA Sequence (5′to 3′)
clathrin heavy chain (CHC)	siCHC	5′-GGAGGGAAGUUACAUAUUATT-3′
caveolin-1	siCav	5′- CCCACUCUUUGAAGCUGUUTT-3′
dynamin II	si dynamin II	5′-GCACUCUGUAUUCUAUUAATT-3′
Rab 5A	siRab5	5′-GCCAGAGGAAGAGGAGTAGACCTTA-3′
Rab 7A	siRab7	5′- UACUGGUUCAUGAGUGAUGUCUUUC- 3′
Rab 9A	siRab9	5′- CCGAGGAUAGGUCAGAUCATT-3′
Rab11A	siRab11	5′- GGGCAAUAAGAGUGAUCUATT-3′
negative control	siNC	5′-UAAUAUGUAACUUCCCUCCTT-3′

**Table 2 viruses-13-01035-t002:** Primer sequences for qRT-PCR.

Name	Primer Sequence (5′to 3′)
BPIV3 F	5′- AACAGAGCGACCCAAAATCAAC -3′
BPIV3 R	5′- ACTTGTCTCCTGATCCCTCTTC -3′
Actin sense	5′- TGACGTGGACATCCGCAAAG -3′
Actin antisense	5′- CTGGAAGGTGGACAGCGAGG -3′

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
