# Peer review of "Bovine Parainfluenza Virus Type 3 (BPIV3) Enters HeLa Cells via Clathrin-Mediated Endocytosis in a Cholesterol- and Dynamin-Dependent Manner"

_viruses, 2021, doi:10.3390/v13061035_

Round 1

Reviewer 1 Report

 Bovine parainfluenza virus type 3 (BPIV3) enters HeLa cells via  clathrin-mediated endocytosis in a cholesterol- and dynamin- dependent manner

Bovine parainfluenza virus 3 (BPIV3) is a crucial causative agent of respiratory disease in young and adult cattle, and yet new antiviral approaches need to be developed. Wei Pan et al., in this article, propose to use the virus internalization pathway as a target for new therapies. To this end, they studied the mechanism of BPIV3 entry into Hela cells using RNA silencing and pharmacological inhibitors.

They investigated clathrin- and caveolae-mediated endocytosis, the role of cholesterol in cell membrane and also the role of micropinocytosis pathway and finally conclude that BPIV3 enters HeLa cells by following a cholesterol-, dynamin-, or clathrin-dependent pathway and requires cathepsins.

More generally, I would say that the authors remain very descriptive without really providing precise mechanistic.

The author showed that the entry of BPIV3 is cholesterol-dependent: these results are not surprising since it has already been shown by Li, L .; Yu, L .and  Hou, X. (Cholesterol-rich lipid rafts play a critical role in bovine parainfluenza virus type 3 (BPIV3) infection. 440 Research in Veterinary Science 2017, 114, 341-347, doi: 10.1016 / j.rvsc.2017.04.009) that the infectivity of BPIV3 was inhibited by MβCD in a dose-dependent manner. They showed that MDBK cells treated by MβCD after virus-entry had no effects on the virus infection, to suggest that BPIV3 infection was associated with lipid rafts in cell membrane during viral entry stage.

The experiments carried out here on HeLa cells gave same conclusions and the author should perhaps consider their results in fig. 1, here more as a control of their HeLa model rather than an "innovative" result.

In this work, the first things that surprises are the use of HeLa cells as a model for BPIV3 infection. This virus is mainly studied on MDBK cells, of the same bovine origin and it would have been appreciated that the human cells tested could be of a tropism closer to this virus. Why not have tested on human epithelial cells type A549? The authors should be clearer about the reasons for their choice.

The results clearly show all the controls carried out (cytotoxicity of the compounds) and the figures are combined each time on the same diagram, which makes it easier to understand the results. However, homogenization must be carried out on the western blots where HN (Fig 1B) and HN BPIV3 can be seen on the others.

The legends of the figures can be improved by adding the abbreviations used in the diagrams (NC, HN for example). The statistical value p is shown in the legend even if no asterisk is shown on the graph.

Why did the authors used sometimes Gapdh and other times b-Actine as control?

Line 152 ; present figure 2B in the text.

Reviewer 2 Report

The study was devoted to the design of experiments to investigate how BPIV3 enters HeLa cells. Reported results demonstrate that BPIV3 enters HeLa cells through clathrin-366 mediated endocytosis (CME) and requires dynamin, cholesterol, and cathepsins.

Overall, the manuscript is well written. there are a few issues that need to be addressed:

1) Results and discussion related to Fig. 5B:
The images shown do not prove internalization without further information regarding:
Which microscope was used to obtain these images? 
What is the blue color showing? 
What is the purple color showing?
z-stacks need to be taken, if possible, to show for sure that internalization is occurring.
Scale bar is missing.
Image quality needs to be improved. The referee suggests presenting larger images, possibly making Fig. 5B its own Figure.

2) Conclusion Section 5 needs to be expanded on and should include a brief summary of the paper's main points. How does it all fit together?

3) Other minor correction suggestions are noted in the attached pdf.

Round 2

Reviewer 1 Report

Response to Reviewer 1 Comments

Point 1: The author showed that the entry of BPIV3 is cholesterol-dependent: these results are not surprising since it has already been shown by Li, L .; Yu, L .and  Hou, X. (Cholesterol-rich lipid rafts play a critical role in bovine parainfluenza virus type 3 (BPIV3) infection. 440 Research in Veterinary Science 2017, 114, 341-347, doi: 10.1016 / j.rvsc.2017.04.009) that the infectivity of BPIV3 was inhibited by MβCD in a dose-dependent manner. They showed that MDBK cells treated by MβCD after virus-entry had no effects on the virus infection, to suggest that BPIV3 infection was associated with lipid rafts in cell membrane during viral entry stage. The experiments carried out here on HeLa cells gave same conclusions and the author should perhaps consider their results in fig.1, here more as a control of their HeLa model rather than an "innovative" result.

Response 1 : Thank you for your professional advice. We agree with your comment. According to your suggestion, the results in Figure 1 and 2 have been put together and generated the new Figure 1, and the Results section and Figure Legend in manuscript has been rewrited .

Comments: The authors moderate their results by compiling them into a single figure. They could have added a comment on these results in the discussion when they mention reference 26

Point 2: In this work, the first things that surprises are the use of HeLa cells as a model for BPIV3 infection. This virus is mainly studied on MDBK cells, of the same bovine origin and it would have been appreciated that the human cells tested could be of a tropism closer to this virus. Why not have tested on human epithelial cells type A549? The authors should be clearer about the reasons for their choice.

Response 2: Thank you for your pointing it out. Here are some reasons for use of HeLa cells as a model for BPIV3 infection. Firstly, the BPIV3 strain used in this study could only be propagated in Hela and MDBK cells so far. Secondly, we have also investigated the entry mechanism of BPIV3 on MDBK cells. We found that BPIV3 utilized different entry mechanism to enter MDBK cells, and these will be submit to the next manuscript.

Comments: I understand the reasoning but perhaps it would be necessary to give an interest to study an entry mechanism in a cellular model which does not correspond to a natural host or a privileged target of this virus. The fact that the mechanisms can be different between the two cell types shows in fact the importance of using a good model especially if the goal is to have a possible target for the development of an effective antiviral therapy.

Point 3: The results clearly show all the controls carried out (cytotoxicity of the compounds) and the figures are combined each time on the same diagram, which makes it easier to understand the results. However, homogenization must be carried out on the western blots where HN (Fig 1B) and HN BPIV3 can be seen on the others.

Response 3: Thank you for your pointing it out. According to your suggestion, all the “HN”

on western blots have been changed to BPIV3 HN to carry homogenization.

Comments: No comment

Point 4:The legends of the figures can be improved by adding the abbreviations used in the diagrams (NC, HN for example). The statistical value p is shown in the legend even if no asterisk is shown on the graph.

Response 4: Thank you for your professional advice. The abbreviations used in the diagrams have been added in the legends of the figures, and the statistical value p was shown in the legend.

Comments: No comment

Point 5:Why did the authors used sometimes Gapdh and other times b-Actine as control? Line 152 ; present figure 2B in the text.

Response 5: Thank you for pointing it out. When we did the exprements in figure 2, the antibody againstβ-Actin has been used up. As a result of the significant disruption that is being caused by the COVID-19 pandemic, the antibody could not be purchased immmediately. So, we chosed  GAPDH as control in figure 2B. 

 Comments: it's a shame that there is this lack of homogeneity just on one figure, but I understand the difficulty of supply during this period.
